# Performance evaluation of national healthcare systems in the prevention and treatment of non-communicable diseases in sub-Saharan Africa

**Kwadwo Arhin**[1]*, **Disraeli Asante-Darko**[2]

**1** Department of Accounting and Finance, Ghana Institute of Management and Public Administration, Accra, Ghana, **2** Department of Management Science, Ghana Institute of Management and Public Administration, Accra, Ghana

* kwarhin@gimpa.edu.gh, arhinkwadwo@gmail.com

**Data Availability Statement:** The minimal data set underlying the results of the manuscript has been deposited in a publicly open repository with URLs of https://zenodo.org/record/8151509m and DOI of

## Abstract

### Background

Non-communicable diseases (NCDs) remain a major public health concern globally, threatening the achievement of sustainable development goal 3.4 (SDG 3.4), which seeks to reduce premature NCD-related deaths by one-third by 2030. According to the World Health Organization (WHO), improving the efficiency of NCD spending (i.e., maximizing the impact of every dollar spent on NCDs) is one of the strategic approaches for achieving SDG target 3.4. This study aims to assess the efficiency and productivity of NCDs spending in 34 sub-Saharan African (SSA) countries from 2015 to 2019.

### Methods

The study employed the data envelopment analysis (DEA) double-bootstrap truncated and Tobit regressions, one-stage stochastic frontier analysis (SFA) model, the Malmquist productivity index (MPI), and spatial autocorrelation analysis to estimate NCDs spending efficiency, identify the context-specific environmental factors that influence NCDs spending efficiency, evaluate total productivity change and identify its components, and assess the spatial interdependence of the efficiency scores.

### Results

The estimated average DEA bias-corrected NCD spending efficiency score was 87.3% (95% CI: 86.2–88.5). Additionally, smoking per capita, solid fuel pollution, alcohol use, governance quality, urbanization, GDP per capita, external funding for NCDs, and private domestic funding for NCDs healthcare services were found to be significantly associated with NCDs spending efficiency. The study also revealed a decline of 3.2% in the MPI, driven by a 10.6% technical regress. Although all countries registered growth in efficiency, except for the Central Africa Republic and DR Congo, the growth in efficiency was overshadowed by the decline in technical change. Global Moran's I test indicated the existence of

https://doi.org/10.5281/zenodo.8151509. They have also been included within the paper and its Supporting information files.

**Funding:** The authors received no specific funding for this work.

**Competing interests:** The authors have declared that no competing interests exist.

significant positive spatial autocorrelation in the efficiency of NCDs spending across SSA countries.

## Conclusion

The study underscores the importance of efficient use of resources in NCDs treatment and prevention and increased investment in NCDs research and development in achieving the SDG target 3.4.

## Introduction

Mortality and morbidity from non-communicable diseases (NCDs) continue to dominate the world's health landscape. Worldwide, non-communicable diseases (NCDs) account for 74% of all deaths, causing 41 million deaths every year. More than 15 million of these deaths occur among people between the ages of 30 and 69, defined as premature deaths [1, 2]. According to estimates, 85% of premature deaths occur in low- and lower-middle-income countries (LLMICs). Evidence shows that through actions that are affordable for every nation, up to 80% of NCDs that cause these premature deaths can be avoided or postponed into old age [3].

In sub-Saharan Africa (SSA), NCDs were responsible for an estimated 37% of deaths in 2019, an increase from 24% in 2000 [4]. NCDs are increasing rapidly in SSA, with a projection that by 2030, NCDs will account for almost three-quarters of all deaths in the region, overtaking communicable, maternal, neonatal, and nutritional (CMNN) diseases combined as the leading cause of mortality and morbidity [2]. This is due to a combination of factors, including changing demographics, increasing urbanization, and changing lifestyles. The major risk factors for NCDs in sub-Saharan Africa (SSA) include unhealthy diets, physical inactivity, tobacco use, harmful use of alcohol, and air pollution [2–4]. According to the Global Burden of Disease Study, NCDs contribute to a significant disability-adjusted life years (DALYs) burden in sub-Saharan Africa (SSA). NCDs accounted for 57% of total DALYs in the region in 2019. The four leading causes of NCD-related DALYs in sub-Saharan Africa (SSA) are cardiovascular diseases, cancer, chronic respiratory disease, and diabetes [5, 6].

The burden of NCD-related DALYs significantly impacts economic development in sub-Saharan Africa (SSA), as it reduces the productive capacity of the workforce and increases healthcare costs. The economic cost of NCDs to the economies of SSA is significant and has been estimated to be in the billions of dollars. This cost includes the direct costs of healthcare and the indirect costs associated with lost productivity and premature mortality [2, 7, 8]. In the context of constrained resources and weak healthcare systems that are still facing the overwhelming burden of CMNN diseases, achieving the Sustainable Development Goal (SDG) 3.4 to reduce premature deaths from NCDs by one-third by 2030 is extremely challenging [7, 9]. The increasing burden of NCDs coupled with the limited available resources to fight NCDs in SSA highlights the importance of achieving value for money in NCD spending. According to WHO, improving the efficiency of NCD spending (i.e., maximizing the impact of every dollar spent on NCDs) is one of the strategic approaches for achieving SDG target 3.4 [10].

The measurement of efficiency in spending is one of the most important components of performance evaluation. In performance evaluation, individuals, groups, organizations, countries, or systems responsible for producing goods or providing services are assessed against a set of goals or standards. This is done to provide feedback for future improvement [11, 12]. Efforts to reduce inefficiency associated with the prevention and treatment of NCDs are

needed to lessen the burden and mitigate the impact on health and economic development in the region. An international benchmarking study of health systems is very important for evaluating the overall performance of health systems concerning NCDs spending efficiency.

However, studies on the efficiency of NCDs spending are limited. Previous studies have analyzed malaria spending efficiency [13, 14], HIV/AIDS spending efficiency [15], the efficiency of tuberculosis spending [16], the efficiency in curbing the COVID-19 pandemic [17, 18], and the efficiency of health systems in general [19–22]. From the extant literature and to the best of our knowledge, only one country-level analysis of efficiency of NCDs spending has been carried out up to date. This analysis was done for 31 provinces of mainland China for the period 2008–2015 [23]. From the accessible literature, no multi-country performance evaluation study has been undertaken on the efficiency and productivity of NCDs spending in SSA. This study aims to fill this gap and build on the existing literature by answering the following questions: (i) How efficient are SSA healthcare systems in their NCDs spending? (ii) What environmental context-specific factors explain differences in NCDs spending efficiency levels across SSA countries? and (iii) What are the causes of NCDs spending efficiency changes?

## Methods and materials

### Study design and analytical models

In the field of healthcare studies, evaluating the efficiency performance of decision-making units (DMUs) is commonly conducted using two major methods: data envelopment analysis (DEA), a widely used non-parametric approach, and stochastic frontier analysis (SFA), a commonly employed parametric method [19, 24, 25]. This study employs both DEA and SFA models to examine the technical efficiency and its determinants of NCDs spending. Additionally, the study uses the Malmquist productivity index to evaluate productivity changes of the health systems and spatial autocorrelation analysis was conducted to assess the clustering or dispersal patterns of NCDs spending efficiency within the geographical confines of SSA.

**Data envelopment analysis (DEA) model.** The current approach for measuring the relative efficiency of decision-making units (DMUs) that use multiple inputs to produce multiple outputs was first introduced by Farrell [26], based on the earlier works of Debreu [27] and Koopmans [28]. Following Farrell's study, Charnes, Cooper, and Rhodes (CCR) proposed the data envelopment analysis (DEA) model [29] which assumes constant returns to scale in production. Banker, Charnes, and Cooper (BCC) extended the CCR model by proposing a production model that assumes variable returns to scale (VRS) which is more flexible [30–34]. The VRS output-oriented DEA approach for estimating the efficiency scores ($\theta_i$) of the DMUs can be obtained by solving the following linear programming model:

$$
\begin{aligned}
&Max\theta, \; \lambda\theta_i \\
Subject\ to \quad &-\boldsymbol{\theta}_i\boldsymbol{y}_i + \boldsymbol{Y}\lambda \geq \mathbf{0}, \\
&\boldsymbol{x}_i - \boldsymbol{X}\lambda \geq \mathbf{0} \\
&\boldsymbol{N1'}\lambda = \boldsymbol{1}\lambda \geq \mathbf{0}
\end{aligned}
\tag{1}
$$

where $y_i$ and $x_i$ denote vectors of the output and input variables, respectively, for the $i^{th}$ country. The output matrix $Y$ has dimensions ($p \times n$), and the input matrix $X$ has dimensions ($q \times n$), where $p$ and $q$ represent the number of output and input variables, respectively, and $n$ represents the number of countries. The value of $\theta_i$, which represents the Shephard output-oriented efficiency score under VRS, ranges from zero to one with a higher score indicating greater efficiency. The weight vector $\lambda$, which has dimensions ($n \times 1$), is used to determine the location of an inefficient country relative to the efficient frontier.

To be able to account for the environmental or external factors that explain the differences in efficiency scores of DMUs, we employed the Simar and Wilson two-stage DEA model [35]. The environmental variables, though affect the prevention and treatment of NCDs, are not under the control of the managers of the NCDs programs and can vary across different health systems. In the first stage, a parametric bootstrap procedure is applied to solve the linear programming problem in Eq 1 to obtain a bias-corrected efficiency score $(\hat{\theta}_i)$ as an estimate for $\theta_i$ in Eq 1. In the second stage, the estimated bias-corrected efficiency scores are used as a dependent variable in a bootstrap regression model to assess the impact of the environmental variables on the efficiency scores as shown in Eq 2.

$$\theta_i = z_i\beta + \varepsilon_i \tag{2}$$

Where $\hat{\theta}_i$ (the bias-corrected technical efficiency score) is estimated by solving Eq 1; $z_i$ is the vector of the environmental variables; $\beta$ is the vector of parameters to be estimated; and $\varepsilon_i$ is truncated normal random variable $(0, \sigma_\varepsilon^2)$. This study adopts Algorithm #2 of the two-stage double bootstrap approach as recommended by Simar and Wilson to estimate Eq 2 [32, 35].

**Stochastic frontier analysis (SFA) model.**   A stochastic frontier production model with time-varying inefficiency is applied to the panel data. The SFA translog production model, as defined in Eq 3. is adopted in this study due to its flexibility in accommodating different production functional forms without the need for their a priori specifications (Hollingsworth, 2008).

$$lnY_{it} = \beta_0 + \sum_{i=1}^{n} \beta_i lnX_{it} + \frac{1}{2}\sum_{i=1}^{n} \beta_{ii} lnX_{it}^2 + \sum_{i=1}^{n}\sum_{j\neq i=1}^{n} \beta_{ij} lnX_i lnX_j + v_{it} - u_{it} \tag{3}$$

Where $Y_{it}$ is the output variable; $X_{it}$ denotes the input variables; $\beta_0$, $\beta_i$, $\beta_{ii}$, and $\beta_{ij}$ represent unknown parameters to be estimated; $v_{it}$ is the random symmetric component of the error term which is assumed to be independent and identically distributed with normal distribution of zero mean and constant variance $(\sigma_v^2)$; and $u_{it}$ represents the non-negative technical inefficiency estimated via Jondrow et al. [36] approach with the mean $z_{it}'\delta$ and variance $\sigma_u^2$. This study adopts the single-step procedure which accounts for the exogenous influences on inefficiency by parameterizing the distribution function of the $u_{it}$ as a function of $z_{it}$ [37–39]. Thus, the inefficiency function for country i at time $t$ is estimated in Eq 4 as follows:

$$u_{it} = z_{it}'\delta + \omega_{it} \tag{4}$$

Where $z_{it}$ is a set of explanatory variables, $\delta$ is a vector of parameters to be estimated, $\omega_{it}$ is the random variable defined by the truncation of the normal distribution with a zero mean and variance $\sigma_u^2$, such that the point of truncation occurs at $-z_{it}'\delta$. Thus, parameters $\delta$ show how variables $z_{it}$ influence the inefficiency term $u_{it}$. If a coefficient is positive, then the corresponding variable is contributing to inefficiency, and if is negative, then the variable and the inefficiency term are inversely related. The parameters of the production function $(\beta)$ and those in the inefficiency component $(\delta)$ are simultaneously estimated by maximum likelihood to ensure a robust analysis of the factors affecting inefficiency [37]. The DEA and SFA models were estimated using Stata version 17.

**Malmquist productivity index.**   The Malmquist productivity index (MPI) is used to measure the total factor productivity of DMUs that use multiple inputs to produce multiple outputs when it is necessary to evaluate their performance over time. Malmquist [40] originally suggested the concept of MPI as a quantity index to analyze the consumer theory of inputs. Caves et al. [41] later advanced the idea and used it in productivity measurement. Then Fare

et al. [42] combined the ideas from Farrell [26] efficiency measurement and Caves et al. [41] productivity measurement to develop the present DEA-based MPI.

The MPI uses DEA to compute the total factor productivity change between two time periods by decomposing it into two: efficiency change and technical change [34, 43]. The efficiency change, denoted as the *catching-up* effect, measures the extent to which a DMU has improved its efficiency relative to its peers over time. However, the technical change, interpreted as the *frontier shift* effect, measures the extent to which a DMU has improved its production technology over time. The output-oriented MPI between two time periods, $t$ and $t + 1$, using period $t$ and period $t + 1$ technology, respectively, is defined as geometric mean of the two periods as given in Eq 5.

$$M_o = (y^{t+1}, x^{t+1}, y^t, x^t) = \left[\frac{D_o^{t+1}(x^{t+1}, y^{t+1})}{D_o^t(x^t, y^t)}\right] \times \left[\frac{D_o^t(x^{t+1}, y^{t+1})}{D_o^{t+1}(x^{t+1}, y^{t+1})} \times \frac{D_o^t(x^t, y^t)}{D_o^{t+1}(x^t, y^t)}\right]^{\frac{1}{2}} \quad (5)$$

Eq 5 shows the calculation of the output-oriented Malmquist Productivity Index (MPI) using distance functions. The variables used are $M_o$ for MPI, $D_o$ for the distance function, $D_o^t(x^t, y^t)$ for the output distance function that measures period $t$ data relative to technology in period $t$, $D_o^t(x^{t+1}, y^{t+1})$ for the output distance function that measures period $t + 1$ data relative to technology in period $t$, $D_o^{t+1}(x^t, y^t)$ for the output distance function that measures period $t$ data relative to technology in period $t + 1$, and $D_o^{t+1}(x^{t+1}, y^{t+1})$ for the output distance function that measures period $t + 1$ data relative to technology in period $t + 1$.

The first term at the right-hand side of Eq 5 in the square bracket, $\left[\frac{D_o^{t+1}(x^{t+1}, y^{t+1})}{D_o^t(x^t, y^t)}\right]$, represents the efficiency change (EFFCH) which indicates whether a DMU is getting closer to or moving farther away from the frontier over time. That is, it measures the improvement or deterioration in the technical efficiency of a DMU over time. It is the ratio of the output-oriented technical efficiency between period $t$ and period $t + 1$. If the value of EFFCH is greater than 1, it means a DMU has become more efficient in period $t + 1$ as compared to period $t$ while a value less than 1 means that the DMU has become less efficient. An EFFCH value of 1 signifies stagnation in efficiency.

On the other hand, the second term of Eq 5, $\left[\frac{D_o^t(x^{t+1}, y^{t+1})}{D_o^{t+1}(x^{t+1}, y^{t+1})} \times \frac{D_o^t(x^t, y^t)}{D_o^{t+1}(x^t, y^t)}\right]^{\frac{1}{2}}$ denotes the geometric mean of the two ratios inside the square bracket. It measures a shift of the frontier or technology (i.e., technical change (TECH)). That is, it captures the technical progress or decline of a DMU. If the value of TECH is greater than 1, it means a positive shift of the frontier (technical progress) between periods $t$ and $t + 1$. A value less than 1 indicates technical regress (a negative shift of the frontier) while a value equal to 1 implies no technical change between periods $t$ and $t + 1$.

The product of efficiency change (EFFCH) and technical change (TECH) measures the Malmquist Productivity Index (MPI) [i.e., $MPI = EFFCH \times TECH$]. MPI value greater than 1 shows growth in productivity while a value less than 1 indicates a decline in productivity between periods $t$ and $t + 1$. An MPI value of 1 means stagnation in productivity.

The efficiency change (EFFCH) component of the MPI can further be decomposed into pure efficiency change (PECH) and scale efficiency change (SECH) by solving two additional linear programming problems under the variable returns to scale (VRS) [43, 44]. The PECH measures the change in the efficiency of a DMU resulting from an improvement or deterioration in the use of its inputs while holding the scale of operations constant. That is, it measures the ability of a DMU to produce more outputs with the same set of inputs. However, SECH measures the change in the efficiency of a DMU resulting from a change in its scale of

operations while holding the efficiency in the use of its inputs constant. In other words, SECH measures the ability of a DMU to operate at a larger or smaller scale with the same level of efficiency in the use of its inputs. The computation of the Malmquist Productivity Index (MPI) and its decomposition into efficiency change (EFFCH) and technical change (TECH) was carried out using DEAP 2.1 software which was developed by Tim Coelli [45]. The EFFCH component was further decomposed into pure efficiency change (PECH) and scale efficiency change (SECH).

**Spatial autocorrelation analysis.**   In this study, we utilized Exploratory Spatial Data Analysis (ESDA) techniques to investigate the relationship between NCDs spending efficiency in individual countries and the corresponding values of this variable in neighboring countries [46, 47]. Our aim was to assess the clustering or dispersal patterns of NCDs spending efficiency within the geographical confines of SSA. Both global spatial autocorrelation analysis and local spatial autocorrelation analysis were employed [47]. Specifically, we conducted global Moran's I test to explore the overall spatial correlation in NCDs spending efficiency across the sampled countries. Additionally, local Moran's I test was applied to delve into the internal correlations, examining how neighboring countries' NCDs spending efficiencies related to each other within specific regions of SSA. These analyses provide valuable insights into the spatial dynamics of NCDs spending efficiency, shedding light on patterns of aggregation and dispersion within the SSA geographical space. The global Moran's I is defined in Eq 6, and the local Moran's I is defined in Eq 7 as follows [46, 48]:

$$I_{global} = \frac{n \sum_{i=1}^{n} \sum_{j=1}^{n} w_{ij} (\theta_i - \overline{\theta})(\theta_j - \overline{\theta})}{\sum_{i=1}^{n} \sum_{j=1}^{n} w_{ij} \sum_{i=1}^{n} (\theta_i - \overline{\theta})^2} \qquad (6)$$

$$I_{local} = \frac{n(\theta_i - \overline{\theta}) \sum_{j=1}^{n} w_{ij}(\theta_j - \overline{\theta})}{\sum_{i=1}^{n} (\theta_i - \overline{\theta})^2} \qquad (7)$$

Where $n$ is the number of sampled spatial units (observations); $\theta_i$ and $\theta_j$ are the NCDs spending efficiency values for countries i and $j$, respectively; $\overline{\theta}$ is the average NCDs spending efficiency; and $w_{ij}$ is the spatial weight matrix which measures the strength of the spatial relationship between countries i and $j$. The Queen contiguity spatial adjacency method was adopted in this study. The global Moran's I value ranges between +1 and −1, with $I > 0$ indicating positive spatial correlation in the NCDs spending efficiency, $I < 0$ denoting negative spatial correlation, and $I = 0$ signaling no spatial correlation. The statistical significance of global Moran's I was determined by the pseudo p-value which was generated through randomization with 999 permutations of the data [47, 48].

The global Moran's I test provides the overall spatial correlation without indicating where the clusters are located or what type of spatial autocorrelation exists [46]. Thus, the local indicator of spatial autocorrelation (LISA), as defined in Eq 7, was applied to examine the degree of spatial correlation between NCDs spending efficiency of a country and its neighboring nations. A positive $I_{local}$ indicates a similarity between a country and its neighbors concerning NCDs spending efficiency, signifying a spatial cluster of similar values. Conversely, a negative $I_{local}$ suggests dissimilarity, highlighting areas of divergence in NCDs spending efficiency [48].

The LISA analysis produces two maps: one showing the statistical significance of the local clusters (i.e., LISA Significance Map), and the other showing the distribution of potential local spatial outcomes in four quadrants–(i) "*High-High*" indicates higher values surrounded by neighboring units with higher values, which means positive spatial autocorrelation; (ii)

"*Low-High*" indicates low values adjacent to neighboring units with higher values, which means negative spatial autocorrelation; (iii) "*Low-Low*" shows lower values surrounded by neighboring units with lower values, which means positive spatial autocorrelation; and (iv) "*High-Low*" indicates higher values adjacent to neighboring units with lower values, which means negative spatial autocorrelation [49]. Units with no spatial autocorrelation are denoted with 'Not Significant'. All the spatial econometric analyses were performed using GeoDa 1.8 software [50].

## Data and data sources

Data related to healthcare systems in 34 sub-Saharan Africa (SSA) countries covering the period from 2015 to 2019 were used in this study. The period and countries sampled for the analysis were based on the availability of data. The data were sourced from the World Health Organization's Global Health Expenditure Database (WHO-GHED) [51] and Global Health Observatory (WHO-GHO) [52], Institute of Health Metrics and Evaluation (IHME) database [53], and the World Bank's World Development Indicators (WB-WDI) [54] and World Governance Indicators (WB-WGI) [55]. The variables used in this study were categorized into input and output variables for the DEA, SFA, and MPI models and environmental variables for the second-stage analysis of the DEA and SFA models.

**Input and output variables.** Since the objective of this study was to evaluate the efficiency and productivity performance of health systems in the treatment and prevention of NCDs, the selection of the input and output variables was based on the economic theory of the production of health, previous similar empirical studies, and availability of relevant data related to NCDs. Table 1 presents the definitions and significance of the input and output indicators used in this study.

We adjusted the two negative output indicators, the NCDs mortality rate (NMR) and NCDs disability-adjusted life years (DALYs) since the DEA framework requires that outputs are measured in such a way that more is preferable. This adjustment was done to ensure that a country with the lowest NCDs-related mortality rate and DALYs will receive the highest scores. Thus, the adjusted NMR (ANMR) and the adjusted DALYs (ADALY) were respectively

**Table 1. Input and output indicators for measuring NCDs spending efficiency.**

| Type | Indicator | Definition | Significance |
|---|---|---|---|
| Input | *Spending* [a] | Expenditure on NCDs per capita measured in power parity purchasing (PPP) rate. | It reflects the level of monetary investment in the prevention and treatment of NCDs. |
| | *Labor* [b] | The number of health workers per 10,000 population. It includes physicians, clinical workers, and community health workers. | It reflects the level of investment in health human resources in each country. |
| Output | *Mortality* [b] | The number of deaths per 100,000 people attributed to NCDs. | It measures impact and prevalence of NCDs on the population. |
| | *DALYs* [b] | The sum of the years lost due to premature death before age 70 years and the number of years lived with disability caused by NCDs per 100,000 people. | It measures the burden of NCDs on the population in each country. |
| | *UHC* [c] | UHC sub-category related to NCDs which is computed using the level of health service coverage, financial protection, and equity in accessing NCDs medical services. | It monitors accessibility, quality, and affordability of healthcare services for prevention, treatment, and management of NCDs. |

Data Source:

[a] WHO-GHED;

[b] IHME;

[c] WHO-GHO

calculated as follows:

$$ANMR = \frac{100,000 - NMR}{NMR} \times 100 \tag{8}$$

$$ADALY = \frac{100,000 - DALYs}{DALYs} \times 100 \tag{9}$$

**Environmental variables.** Some environmental variables were chosen to assess their relationship with NCDs spending efficiency in the second stage of the DEA model. These variables were selected based on the theoretical and empirical literature, association with NCDs, and data availability. Table 2 presents the definitions of the variables and their expected effect on the efficiency of NCDs spending.

Based on these environmental variables, the second-stage bootstrap regression in Eq 2 and the SFA inefficiency model in Eq 4 are empirically modeled as follows:

$$\hat{\theta}_{it} = \beta_0 + \beta_1 Smok_{it} + \beta_2 Alc_{it} + \beta_3 Pull_{it} + \beta_4 Urb_{it} +$$
$$\beta_5 InGDPpc_i + \beta_6 Gov_{it} + \beta_7 Ext_{it} + \beta_8 lnPriv_{it} + \varepsilon_{it} \tag{10}$$

where $\theta_{it}$ is the technical (in)efficiency score for country $i$ at time $t$; $Smok_{it}$ is smoking per capita; $Alc_{it}$ is alcohol use per capita; $Pull_{it}$ is pollution from solid fuels; $Urb_{it}$ is the proportion of the population living in urban centers; $InGDPpc_{it}$ is the log of GDP per capita at purchasing power rate; $Gov_{it}$ denotes governance quality; $Ext_{it}$ is external funding for NCDs as share of

**Table 2. Definitions and justifications for the inclusion of environmental variables.**

| Variable Name | Variable Definition and Justification (Expected effect on efficiency) | Data Source |
|---|---|---|
| Smoking | The average number of cigarettes smoked by people aged 15 years and above in a given population in a year. It is used to measure the prevalence of smoking. Scientific and medical evidence indicates that cigarette smoking causes NCDs such as chronic respiratory diseases [56] (−). | IHME |
| Alcohol use | The average volume of pure alcohol, expressed in liters, consumed by people aged 15 years and above per year. Many studies have shown a significant relationship between alcohol consumption and NCDs [57] (−). | IHME |
| Pollution from solid fuel | Households' average exposure to particulate matter measured in micrograms of PM2.5 per cubic meter of air resulting from burning solid fuels such as charcoal or wood for heating, cooking, or other household activities. A study by Faizan and Thakur [58] is among several other studies that reveal a significant association between solid fuel use and NCDs, particularly respiratory diseases (−). | IHME |
| Governance quality | The average of the World Bank's six indicators of governance: voice and accountability, stability and absence of violence, government effectiveness, regulatory quality, rule of law, and control of corruption. The score of each indicator ranges from −2.5 to +2.5 with higher scores indicating better performance. These variables have been previously used in empirical efficiency studies [15] (+). | WB-WGI |
| Urbanization | The proportion of the population living in urban areas. Several pieces of evidence indicate a positive association between urbanization and increased risk of NCDs such as diabetes and hypertension in LLMICs due to reduced physical activity, availability of unhealthy food options, and exposure to pollution [59] (−). | WB-WDI |
| GDP per capita | Per capita Gross Domestic Product (GDP) per year measured in 2019 constant prices was used as a proxy for income. Evidence shows that low income and low education are significantly associated with an increased prevalence of NCDs and multi-morbidity in low-income study settings [59, 60] (+). | WB-WDI |
| NCDs external funding | External funding for NCDs as a proportion of total external health expenditure. Increases in external funding for NCDs reduce the incidence of out-of-pocket payments for healthcare services (+). | WHO-GHED |
| NCDs Private. Domestic funding | Private domestic expenditure on NCDs per capita. A greater private domestic expenditure on NCDs in SSA may signify higher financial barriers to access to NCDs' healthcare services (−). | WHO-GHED |

**Notes:** WHO-GHED = World Health Organization's Global Health Expenditure Database; WB-WDI = World Bank's World Development Indicators; WB-WGI = World Bank's World Governance Indicators; IHME = Institute of Health Metrics and Evaluation Database.

total external health funding; $lnPriv_{it}$ is the log of private domestic expenditure per capita on NCDs; and $\varepsilon_{it}$ is the error term.

## Results

### Descriptive statistics

Table 3 presents a summary of descriptive statistics of the input and output variables for the 34 sub-Saharan African (SSA) countries covering the five years of the study from 2015 to 2019. Per capita expenditure on NCDs measured at the international dollar purchasing power parity (PPP) rate increased from an average of $106.67 to $122.49, representing a growth of 3.70% per annum. However, we observed a significant heterogeneity across the countries which tends to increase slightly over the years. Mauritius recorded the highest per capita expenditures on NCDs throughout the period while Mozambique registered the lowest values. For the health workers' density, we did not witness any significant change during the study period. It increased from an average of 5.05 to 5.56 per 10,000 population from 2015 to 2019, representing a 2.9% annual growth.

On the output side, the NCDs-related mortality rate decreased from an average of 652.60 per 100,000 people in 2015 to 625.20 in 2019, representing an annual decline of 1.05%. The

**Table 3. Descriptive statistics of the input and output variables (2015–2019).**

|  | 2015 | 2016 | 2017 | 2018 | 2019 |
|---|---|---|---|---|---|
| *Inputs* |  |  |  |  |  |
| NCDs Spending per capita |  |  |  |  |  |
| Mean | 106.67 | 110.75 | 110.56 | 118.08 | 122.49 |
| Standard Deviation | 202.04 | 211.22 | 212.75 | 226.84 | 239.97 |
| Minimum | 0.92 | 0.96 | 0.92 | 0.98 | 0.96 |
| Maximum | 810.90 | 819.00 | 859.76 | 921.02 | 1017.96 |
| Health Workers Density |  |  |  |  |  |
| Mean | 5.05 | 5.21 | 5.35 | 5.49 | 5.63 |
| Standard Deviation | 3.25 | 3.41 | 3.56 | 3.65 | 3.71 |
| Minimum | 1.35 | 1.41 | 1.45 | 1.49 | 1.52 |
| Maximum | 15.06 | 15.91 | 16.62 | 17.15 | 17.44 |
| *Outputs* |  |  |  |  |  |
| NCDs Mortality Rate |  |  |  |  |  |
| Mean | 652.60 | 645.17 | 637.72 | 631.54 | 625.20 |
| Standard Deviation | 120.20 | 113.95 | 109.71 | 101.67 | 99.49 |
| Minimum | 436.00 | 450.90 | 454.01 | 472.90 | 476.19 |
| Maximum | 995.32 | 969.27 | 943.31 | 924.21 | 917.09 |
| NCDs DALYs |  |  |  |  |  |
| Mean | 23434.04 | 23314.93 | 23194.87 | 23068.95 | 22896.02 |
| Standard Deviation | 2429.96 | 2376.19 | 2342.93 | 2283.04 | 2266.54 |
| Minimum | 18673.79 | 18887.88 | 18968.49 | 18823.53 | 18648.37 |
| Maximum | 30632.56 | 30401.61 | 30216.96 | 30011.11 | 29741.54 |
| UHC on NCDs |  |  |  |  |  |
| Mean | 61.24 | 61.79 | 63.41 | 64.12 | 64.94 |
| Standard Deviation | 7.52 | 8.06 | 8.52 | 8.64 | 8.69 |
| Minimum | 40.00 | 41.00 | 43.00 | 44.00 | 45.00 |
| Maximum | 79.00 | 79.00 | 79.00 | 79.00 | 80.00 |

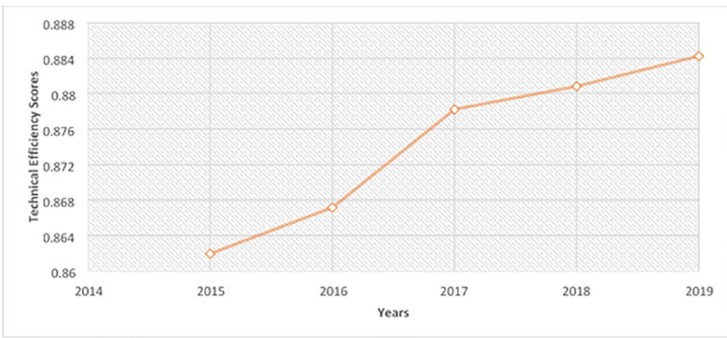

**Fig 1. DEA bias-corrected average technical efficiency scores.**

NCDs-related disability-adjusted life years (DALYs) witnessed a similar trend but with a much slower pace of 0.57% decline per annum. While the Central Africa Republic and Eswatini recorded the highest values for NCDs-related mortality rates and disability-adjusted life years (DALYs), Mauritania and Tanzania were among the countries with the lowest values.

The UHC service coverage sub-index on NCDs measures the degree to which a country's health system is providing the needed services to treat and prevent NCDs. It is measured on a scale of 0 to 100, with 100 indicating full-service coverage. The average UHC service coverage sub-index on NCDs increased from 61.24 in 2015 to 64.94 in 2019, registering a paltry 1.51% increase per annum. The cross-country variations in the UHC service coverage sub-index on NCDs were quite revealing, ranging from a minimum of 40 (in Seychelles in 2015) to a maximum of 80 (in Ethiopia in 2019) (S1 Appendix).

### Results of the DEA model

Fig 1 shows a summary of average bias-corrected efficiency scores estimated using the output-oriented variable returns to scale (VRS) DEA model across 34 national health systems in SSA from 2015 to 2019. Detailed estimates of each national health system and their rankings are presented in S2 Appendix. Fig 1 shows that the efficiency scores have been increasing throughout the study period. The highest increase in average efficiency score was witnessed between the 2016 and 2017 periods from 0.867 to 0.878.

Table 4 shows that the average NCDs spending technical efficiency is 87.34%, indicating that a potential savings of 12.66% of NCDs spending per capita to achieve the same level of NCDs health outcomes if all the national health systems were to be performing as efficiently as their best-performing peers. The results suggest that, on average, low- and lower-middle-income countries were more efficient in their NCDs spending than the upper-middle- and high-income countries.

**Table 4. Average efficiency scores based on the income level of Sub-Saharan African countries.**

| Income Groups | Average | 95% Confidence Interval | Potential Improvement in NCDs Healthcare Output (%) |
|---|---|---|---|
| Low-income | 0.8936 | (0.8752–0.9119) | 10.64 |
| Lower-middle-income | 0.8912 | (0.8772–0.9053) | 10.88 |
| Upper-middle-income | 0.7743 | (0.7509–0.7977) | 22.57 |
| High-income | 0.8158 | (0.8072–0.8244) | 18.42 |
| **Overall Average** | **0.8734** | **(0.8619–0.8849)** | **12.66** |

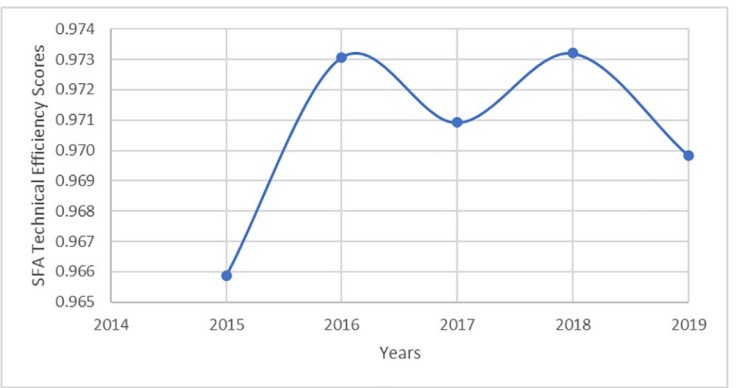

**Fig 2. Average technical efficiency scores based on the SFA model (time dimension).**

## Results of the SFA model

Fig 2 shows that the estimated SFA technical efficiency scores were relatively high and stable over the study period with an average score of 0.971 (95% CI: 0.956–0.985). We found a significant moderate correlation between DEA and SFA efficiency scores ($\rho = 0.534$, $p < 0.0186$). Some specific countries were found to be consistently the most efficient across the analyses, such as Ethiopia (DEA 0.973, SFA 0.999) and Sao Tome & Principe (DEA 0.963, SFA 0.999). Conversely, certain nations were consistently ranked among the least efficient countries, such as the Central Africa Republic (DEA 0.737, SFA 0.725) and Sierra Leone (DEA 0.796, SFA 0.956).

## Effects of environmental variables on NCDs spending efficiency

The DEA bias-corrected technical efficiency scores were regressed against a set of environmental variables using bootstrap regression analysis and the results are presented on Table 5, while Table 6 presents the results of the translog health production function and the regression

**Table 5. Bootstrap DEA regression results[1].**

| Variables [2] | Estimated Coefficient | Standard Error | 95% confidence interval | |
|---|---|---|---|---|
| | | | Lower | Upper |
| Constant | 0.916*** | 0.224 | 0.486 | 1.386 |
| Smoking | -0.919*** | 0.160 | -1.240 | -0.621 |
| Alcohol use | -0.001 | 0.002 | -0.005 | 0.003 |
| Pollution from solid fuel | -0.176*** | 0.069 | -0.319 | -0.044 |
| Governance quality | 0.085*** | 0.019 | 0.050 | 0.123 |
| Urbanization | -0.002*** | 0.001 | -0.003 | -0.001 |
| GDP per capita | -0.082*** | 0.020 | -0.123 | -0.043 |
| NCDs external funding | 0.003** | 0.001 | 0.000 | 0.006 |
| Private domestic funding for NCDs | -0.000** | 0.000 | -0.001 | -0.000 |
| Sigma | 0.073*** | 0.005 | 0.061 | 0.081 |

[1] The coefficients are computed by 2000 bootstrap iterations.

[2] Dependent variable: Bias-corrected efficiency scores (*i.e.* $0 < \hat{\theta}^{**} < 1$).

***and **represent statistical significance at levels 1 and 5%, respectively.

**Table 6. One-step SFA estimation of NCDs health production frontier and inefficiency function.**

| | Estimated | Robust | 95% Confidence Interval | |
|---|---|---|---|---|
| | Coefficient | Std. Error | Lower | Upper |
| *Production Frontier* | | | | |
| Intercept ($\beta_0$) | 5.0011*** | 0.0268 | 4.9487 | 5.0536 |
| Spending ($\beta_1$) | 0.0288** | 0.0118 | 0.0056 | 0.0519 |
| Labour ($\beta_2$) | 0.1210*** | 0.0408 | 0.0410 | 0.2011 |
| Spending × Labour ($\beta_{12}$) | 0.0580*** | 0.0148 | 0.0290 | 0.0871 |
| Spending$^2$ ($\beta_{11}$) | -0.0318** | 0.0041 | -0.0398 | -0.0238 |
| Labour$^2$ ($\beta_{22}$) | -0.0880*** | 0.0342 | -0.1550 | -0.0210 |
| *Inefficiency Function* | | | | |
| Intercept | -12.7299*** | 4.1151 | -20.7954 | -4.6643 |
| Smoking | 4.2227*** | 1.3073 | 1.6603 | 6.7850 |
| Alcohol use | 0.7173*** | 0.1514 | 0.4206 | 1.0140 |
| Pollution from solid fuel | 6.1367*** | 2.0033 | 2.2103 | 10.0631 |
| Governance quality | -1.6686*** | 0.3586 | -2.3714 | -0.9658 |
| Urbanization | 0.0441*** | 0.0163 | 0.0122 | 0.0761 |
| GDP per capita | 0.2558 | 0.1719 | -0.0811 | 0.5927 |
| NCDs external funding | -0.0073 | 0.0215 | -0.0495 | 0.0349 |
| *Model Parameters* | | | | |
| Sigma u ($\sigma_u$) | 0.0379* | 0.0206 | 0.0130 | 0.1100 |
| Sigma v ($\sigma_v$) | 0.0526*** | 0.0031 | 0.0468 | 0.0591 |
| Lambda ($\lambda$) | 0.7198*** | 0.0216 | 0.6775 | 0.7622 |
| Observations | 170 | | | |
| Number of countries | 34 | | | |
| Log-likelihood | 247.6 | | | |
| Wald Chi2 | 282.48 | | | |
| Prob. $> \chi^2$ | 0.0000 | | | |

***, ** and * represent statistical significance at levels 1%, 5%, and 10%, respectively.

of the estimated SFA technical inefficiency scores regressed against the selected environmental variables. We used pair-wise Pearson's correlation coefficients and variance inflation factor (VIF) tests to investigate the presence of multicollinearity in the model (S3 and S4 Appendices). The results indicated no high risk of multicollinearity (the mean-variance inflation factor was 3.26). Again, we applied the robust standard errors in the estimation of the SFA model to correct for any potential heteroscedasticity problem in the data.

According to the results presented in Table 5, smoking, pollution from solid fuel, urbanization, GDP per capital, and NCDs private domestic funding were negatively associated with NCDs spending efficiency scores at 1% level of significance, while good governance and NCDs external funding were positively associated with the NCDs spending technical efficiency scores. The signs of the coefficients were consistent with a priori expectations with the exception of income which was proxied by GDP per capita.

Table 6 presents the results of the one-step estimation of the parameters of the health production frontier ($\beta$) and those in the inefficiency model ($\delta$) by maximum likelihood. The signs of all the coefficients in the health production frontier are consistent with theory. The effects of the NCDs inputs (spending and labour) at their levels ($\beta_1$ and $\beta_2$) have positive significant effects on the outcome variable. However, the coefficient of the quadratic terms ($\beta_{11}$ and $\beta_{22}$)

have negative significant effects on the outcome variable. This implies that the NCDs input variables exhibit diminishing marginal effects on the NCDs outcome variable.

For the inefficiency component, since the inefficiency scores were used as the dependent variable, a negative coefficient for a variable signifies that the variable exerts a negative impact on inefficiency. Put simply, an increase in the variable's value leads to a reduction in inefficiency in NCDs spending. The results, as presented in Table 6, closely align with those derived from the DEA bootstrap regression model (Table 5). This convergence underscores the robustness of the DEA model. Specifically, it was observed that variables such as smoking, alcohol use, solid fuel-related pollution, urbanization, and GDP per capita were positively associated with higher NCDs inefficiency scores. In contrast, quality of governance and external funding for NCDs were negatively associated with an increase in inefficiency scores, signifying their positive impact on NCDs spending efficiency.

## Results of the Malmquist productivity index

The averages of the Malmquist productivity index (MPI) and its decomposition at the country level over the period of the study are presented in Table 7. The estimates indicate that, on average, the productivity of healthcare services for NCDs declined by 3.2% between 2015 and 2019 with a total factor productivity change (MPI) score of 0.968. This decline was largely driven by a regress in technology (TECH).

The average technical change (TECH) score was 0.894, an indication of about a 10.6% decline over the five-year period. The decline in technical change (TECH) eclipsed the growth in efficiency change (EFFCH) of 8.2%. The average pure efficiency change (PECH) and scale efficiency change (SECH) showed a growth of about 0.2% and 7.9%, respectively. These results imply that the 8.2% growth in efficiency change (EFFCH) was largely driven by the growth in scale efficiency.

Table 8 summarizes the yearly average indices of the Malmquist total factor productivity (MPI) and its components from 2015 to 2019, with 2015 serving as the reference year. The data reveals that while there was marginal growth in the technical change in the first three years (1.2%, 1%, and 0.8%, respectively), the region experienced an overall decline in the technical change of 10.6%. This decline was largely due to the significant decrease of 37.9% in 2019, which overshadowed the earlier marginal growths. Conversely, the efficiency change (EFFCH) component showed an opposite trend. Although there were marginal declines in 2016 (4.9%), 2017 (5.5%), and 2018 (0.7%), there was a considerable growth of 53.5% in 2019. The data further reveals that total factor productivity change (MPI) only recorded growth in 2018, and the performance of all components varied inconsistently over the years. For instance, while the pure efficiency changes exhibited growth in two years, it registered a decline in the other two years.

Table 9 shows the summary of the indices of MPI and its components across all 34 national health systems sampled for this study. The results show that 15 out of the 34 countries, representing 44%, registered growths in total factor productivity (MPI) between 2015 and 2019. The rest of the 19 countries experienced a decline in the MPI. The top three performing countries were South Africa, Congo, and Malawi while the worst performing countries were Central Africa Republic, DR Congo, and Guinea. On the front of efficiency change (EFFCH), 31 out of the 34 countries experienced a growth while 2 and 1 countries registered a decline and stagnation, respectively.

The results also show that 19 and 31 countries experienced growth in pure efficiency change (PECH) and scale efficiency change (SECH), respectively. All 34 countries registered a decline in technical change (TECH) between 2015 and 2019.

**Table 7. Malmquist productivity index and its sub-components (2015–2019 averages).**

| Country | Efficiency Change | Technical Change | Pure Efficiency Change | Scale Efficiency Change | Malmquist Productivity Index |
|---|---|---|---|---|---|
| Benin | 1.094 | 0.862 | 1.005 | 1.089 | 0.944 |
| Botswana | 1.136 | 0.893 | 1.000 | 1.136 | 1.014 |
| Burkina Faso | 1.138 | 0.885 | 1.013 | 1.123 | 1.007 |
| Cabo Verde | 1.055 | 0.890 | 0.983 | 1.073 | 0.939 |
| Central African Rep. | 0.886 | 0.855 | 0.968 | 0.915 | 0.758 |
| Comoros | 1.125 | 0.863 | 1.007 | 1.118 | 0.971 |
| Congo | 1.121 | 0.975 | 1.013 | 1.107 | 1.093 |
| Côte d'Ivoire | 1.092 | 0.892 | 1.010 | 1.081 | 0.973 |
| DR. Congo | 0.794 | 0.973 | 0.990 | 0.802 | 0.773 |
| Eswatini | 1.148 | 0.874 | 1.002 | 1.146 | 1.003 |
| Ethiopia | 1.139 | 0.882 | 1.000 | 1.139 | 1.005 |
| Gabon | 1.074 | 0.906 | 0.997 | 1.077 | 0.973 |
| Ghana | 1.147 | 0.885 | 1.018 | 1.127 | 1.015 |
| Guinea | 1.032 | 0.884 | 0.980 | 1.053 | 0.912 |
| Kenya | 1.080 | 0.883 | 1.002 | 1.078 | 0.954 |
| Liberia | 1.158 | 0.872 | 1.013 | 1.143 | 1.009 |
| Malawi | 1.174 | 0.868 | 1.026 | 1.144 | 1.019 |
| Mali | 1.053 | 0.894 | 1.008 | 1.045 | 0.942 |
| Mauritania | 1.025 | 0.906 | 1.000 | 1.025 | 0.928 |
| Mauritius | 1.071 | 0.906 | 0.995 | 1.077 | 0.971 |
| Mozambique | 1.000 | 0.932 | 1.000 | 1.000 | 0.932 |
| Namibia | 1.087 | 0.900 | 0.995 | 1.092 | 0.979 |
| Niger | 1.112 | 0.902 | 1.000 | 1.112 | 1.004 |
| Nigeria | 1.146 | 0.883 | 1.005 | 1.140 | 1.012 |
| Sao Tome & Principe | 1.106 | 0.906 | 1.005 | 1.101 | 1.002 |
| Senegal | 1.115 | 0.906 | 1.008 | 1.107 | 1.010 |
| Seychelles | 1.102 | 0.906 | 0.998 | 1.104 | 0.999 |
| Sierra Leone | 1.031 | 0.903 | 1.000 | 1.031 | 0.931 |
| South Africa | 1.172 | 0.883 | 1.015 | 1.154 | 1.035 |
| Tanzania | 1.096 | 0.878 | 1.004 | 1.091 | 0.961 |
| Togo | 1.050 | 0.893 | 1.011 | 1.039 | 0.937 |
| Uganda | 1.151 | 0.872 | 1.012 | 1.138 | 1.004 |
| Zambia | 1.166 | 0.873 | 1.023 | 1.140 | 1.018 |
| Zimbabwe | 1.049 | 0.879 | 0.977 | 1.074 | 0.922 |
| **Average** | **1.082** | **0.894** | **1.002** | **1.079** | **0.968** |

**Table 8. Summary of Malmquist productivity index (longitudinal dimension).**

| Year | Efficiency Change | Technical Change | Pure Efficiency Change | Scale Efficiency Change | Total Factor Productivity Change (MPI) |
|---|---|---|---|---|---|
| 2015–2016 | 0.951 | 1.012 | 0.997 | 0.953 | 0.962 |
| 2016–2017 | 0.945 | 1.010 | 1.013 | 0.933 | 0.955 |
| 2017–2018 | 0.993 | 1.008 | 0.999 | 0.993 | 1.001 |
| 2018–2019 | 1.535 | 0.621 | 1.000 | 1.536 | 0.954 |
| Average (2015–2019) | 1.082 | 0.894 | 1.002 | 1.079 | 0.968 |
| **Growth (+)/Decline (-)** | **+8.2%** | **-10.6%** | **+0.2%** | **+7.9%** | **-3.2%** |

**Table 9. Average productivity performance in the prevention and treatment of NCDs.**

|  | Efficiency Change | Technical Change | Pure Efficiency Change | Scale Efficiency Change | Total Factor Productivity Change (MPI) |
|---|---|---|---|---|---|
| Growth | 31 (91%) | 0 | 19 (56%) | 31 (91%) | 15 (44%) |
| Decline | 2 (6%) | 34 (100%) | 9 (26%) | 2 (6%) | 19 (56%) |
| Stagnation | 1 (3%) | 0 | 6 (18%) | 1 (3%) | 0 |
|  | 34 | 34 | 34 | 34 | 34 |

In order to comprehensively evaluate the performance of the 34 SSA countries, the bias-corrected efficiency scores estimated from the DEA model and the Malmquist productivity indices (TFPCH) were plotted as quadrant distribution. The average efficiency scores (0.874) and average of TFPCH (0.968) were used to divide the scatter plot into four quadrants (Fig 3). It reveals that countries such as Sao Tome & Principe, Ethiopia, Malawi, and Ghana performed above the averages of the two metrics, and can serve as benchmarks. On the other hand, countries such as Central Africa Republic and Democratic Republic of Congo performed below the averages of the two metrics.

## Results of spatial autocorrelation analysis

We computed the global Moran's I using GeoDa 1.8 software for 34 Sub-Saharan African (SSA) countries, utilizing the estimated DEA bias-corrected efficiency scores. The Z-statistical test value and P-value for the global Moran's I were obtained to assess the overall spatial correlation of the NCDs spending efficiency scores, as presented in Table 10. The findings reveal that, throughout the study period, the global Moran's I for NCDs spending efficiency in each country demonstrated a positive value and passed the significance test ($p < 0.05$). This outcome signifies a significant positive spatial autocorrelation in the efficiency of NCDs spending across Sub-Saharan Africa (SSA), indicating that countries with similar efficiency scores tended to cluster together in space. Furthermore, the Moran's I index illustrated a degree of stability, suggesting that the spatial effects on the overall NCDs spending efficiency remained moderately consistent over the course of the study.

Based on the global Moran's I computation, we proceeded to conduct a Local Spatial Autocorrelation Analysis (LISA) to pinpoint the specific locations of clustering. To visualize this,

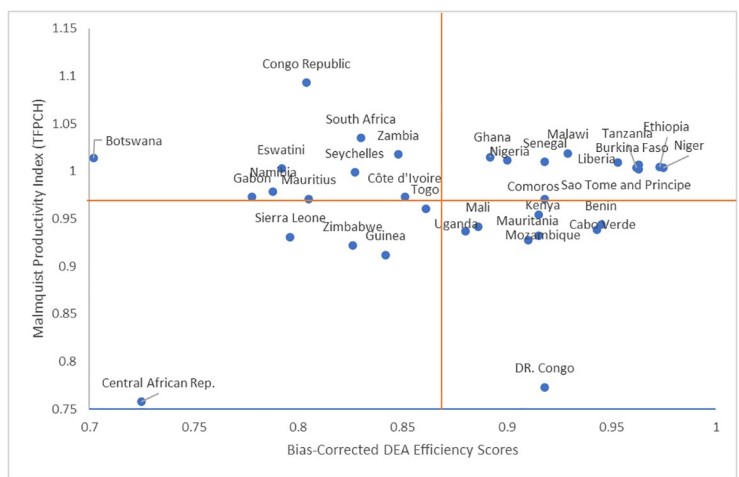

**Fig 3. Scatter plot of DEA efficiency scores and Malmquist productivity index (TFPCH).**

**Table 10. Global Moran's I of NCDs spending efficiency in SSA from 2015 to 2019.**

| Year | Moran's I | Z-statistic | P-value |
|------|-----------|-------------|---------|
| 2015 | 0.385 | 3.0307 | 0.001 |
| 2016 | 0.353 | 2.7887 | 0.003 |
| 2017 | 0.420 | 3.2451 | 0.001 |
| 2018 | 0.322 | 2.5465 | 0.009 |
| 2019 | 0.370 | 2.8680 | 0.004 |

we constructed a Moran's I scatter plot (Fig 4) aimed at identifying the precise areas where local spatial agglomeration of NCDs spending efficiency occurred. The scatter plot reveals that, over the five-year study period, NCDs spending efficiency in SSA exhibited a positive correlation with most countries positioned in the first quadrant (indicating high aggregation) and the third quadrant (indicating low aggregation). In simpler terms, countries with high NCDs spending efficiency tended to cluster with other high-efficiency countries, while low-efficient countries tended to cluster with fellow low-efficient countries. This clustering phenomenon appeared to intensify over time. It is worthy to note that the number of "High-High" (H-H)

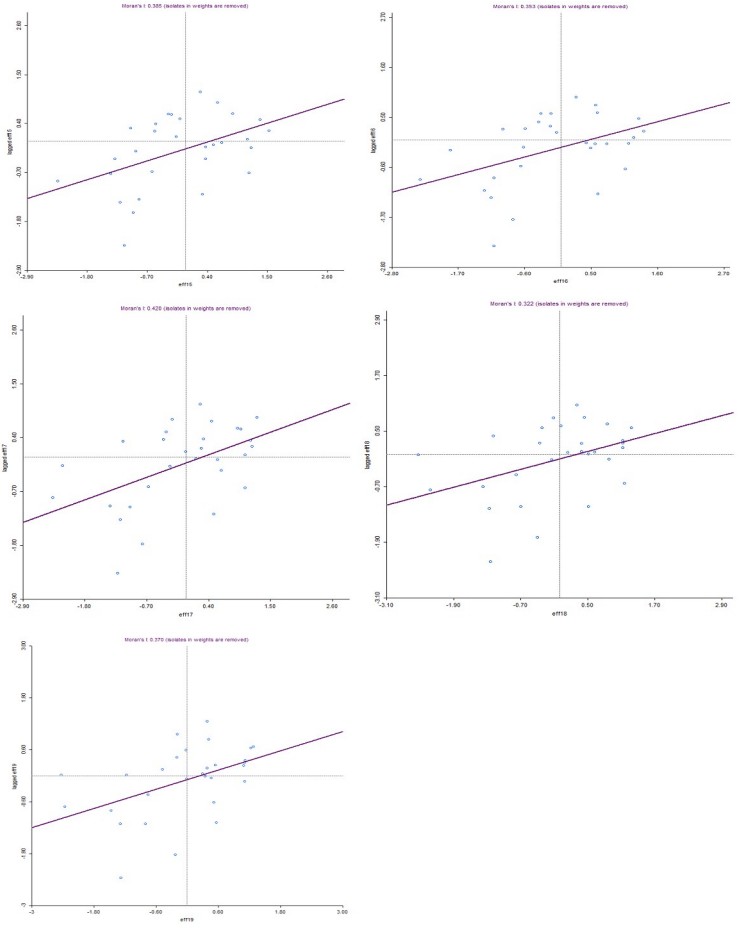

**Fig 4. Moran's I scatter plot.**

clusters increased more rapidly over time compared to "Low-Low" (L-L) clusters. This observation implies that highly efficient countries in SSA exhibited a higher degree of clustering than their less efficient counterparts, which, in turn, had a positive impact on overall efficiency.

## Robustness checks

The efficiency and productivity indices can be sensitive to the number of input and output variables used the DEA model in relation to the number of decision-making units (DMUs). If the number of the DMUs is relatively small, the indices can be overestimated [61]. It is, therefore, suggested that the number of DMUs must at least be three times more than the number of the input and output variables [62]. This constraint is not binding in this current study since the number of DMUs is three times more than the number of the input and output variables.

We conducted robustness checks using different combinations of input and output variables and by excluding countries considered to be potential outliers to test the base model. The most sensitive scenario was when UHC on NCDs was used as the only output variable when the mean efficiency score changed from 0.873 (base model) to 0.806 (Fig 5). The closeness of the mean efficiency scores from all the different scenarios indicates the stability of the base model used in this study.

Again, we used an alternative model, Tobit regression, to test the robustness of the bootstrap regression employed to examine the association between the estimated efficiency scores and the environmental variables. S5 Appendix presents the Tobit regression results. The high degree of similarity of the results compared with that of the bootstrap regression in terms of coefficient values, signs, and statistical significance is an indication of the robustness of the empirical evidence given in this study.

## Discussion

The results of the study revealed that although there has been improvement in the efficiency of resource utilization for the prevention and treatment of NCDs over time, a certain degree of inefficiency remains. The findings from the DEA model suggest that, on average, the technical efficiency of NCDs spending was 87.3% (95% CI: 86.2%–88.5%) over the period, 2015–2019. This implies that given the available resources for the prevention and treatment of NCDs, health systems in SSA could potentially improve NCD-related health outcomes.

It is notable that low- and lower-middle-income countries (LLMICs) demonstrated greater efficiency in their spending on non-communicable diseases (NCDs) compared to upper-

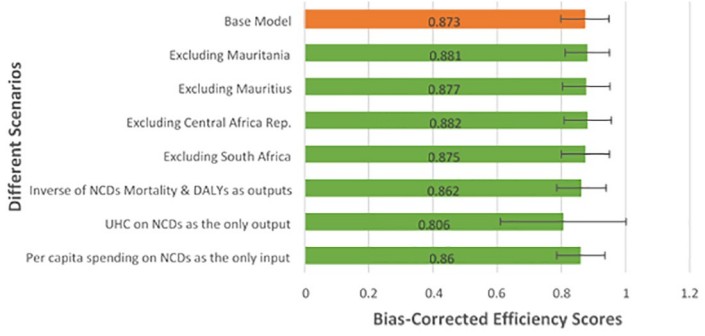

**Fig 5. Average bias-corrected efficiency scores from the robustness checks results.**

middle- and high-income countries (UMHICs), contradicting some earlier studies on health system efficiency that showed better performance by UMHICs [15, 16, 63–65]. Additional analyses of the data revealed more information for the four income groups under consideration (S6 Appendix).

For example, low-income countries (LICs) spent significantly far less on NCDs (mean = $12.61, SD = $8.99) per capita (PPP) than upper-middle-income countries (UMICs) (mean = $384.5, SD = $200). However, NCD-related health outcomes, measured by the NCD-related mortality rate per 100,000 population, were better in LICs (mean = 652.5, SD = 109.3) than in UMICs (mean = 682.2, SD = 54.5). An even more notable difference was observed when universal health coverage (UHC) of NCDs healthcare services was used as the outcome variable. The significant differences in NCD-related health outcomes between LLMICs and UMHICs could be attributed to programs specifically designed for LLMICs to enhance the coverage of essential healthcare services for the prevention and treatment of NCDs. These targeted programs include the World Health Organization's Package of Essential Non-Communicable Diseases (PEN) and the Global Hearts Initiative, which are integrated into the health systems of LLMICs at the primary healthcare level [66, 67].

The study finds that contextual environmental factors significantly influence the efficiency of NCDs spending, which is consistent across all the three models used: DEA double-bootstrap truncated regression (Table 5), SFA inefficiency component model (Table 6), and Tobit regression model (S5 Appendix). The results reveal that lower levels of smoking, alcohol use, and pollution from solid fuels are strongly associated with higher performance of national health systems, in line with previous studies [56, 58]. Based on these findings, clear policy recommendations can be made. Governments should implement tobacco control policies, such as increasing tobacco taxes, enforcing smoke-free laws in public places, and incentivizing smokers to quit. Additionally, public awareness campaigns should be undertaken to educate households and encourage them to switch from solid fuels to cleaner energy sources, such as natural gas and electricity.

The study finds that higher governance quality is a significant factor in a country's NCDs efficiency performance, which is consistent with previous research [15, 16, 33, 64]. Governance has a significant impact on efficiency in several ways. Firstly, governance can affect health inputs such as health spending and health personnel quality. Rajkumar and Swarooop [68] demonstrate that poor governance, characterized by corrupt and ineffective bureaucracy, reduces the effectiveness of public health spending in reducing under-5 mortality rates. Secondly, good governance enhances the capacity of governmental and non-governmental institutions to develop, coordinate, and implement effective policies, resulting in higher returns on health investments [69, 70]. Therefore, it is crucial for governments to prioritize and invest in improving governance quality to increase NCDs spending efficiency performance.

Urbanization was found to be negatively associated with NCDs spending efficiency performance. The negative impact of urbanization on health system performance in the prevention and treatment of NCDs coincides with a number of previous studies that have investigated this relationship and found that urbanization is associated with higher rates of NCDs in LLMICs [71, 72]. This finding from this study highlights the need for policies and interventions that address the unique challenges faced by urban dwellers in preventing and managing NCDs.

Another noteworthy negative association is between income (proxied by GDP per capita) and NCDs spending efficiency performance. The finding that UMHICs have lower efficiency performance in NCDs spending is surprising and goes against previous health system efficiency studies [59, 73, 74]. However, this could be explained by the high-efficiency performance of LLMICs and the lower performance of UMHICs. Policy recommendations should focus on improving the efficiency of NCDs spending in UMHICs in SSA. One approach could

be to prioritize investments in programs and interventions that have been shown to be effective in improving NCD-related outcomes. Additionally, promoting greater collaboration and coordination among healthcare providers and institutions can improve the delivery of NCD-related services and reduce duplication of efforts. Finally, UMHICs should consider investing in strengthening their healthcare systems and addressing any inefficiencies that may be contributing to lower performance in NCD-related outcomes.

In this study, we have observed a significant association between mechanisms for financing NCDs healthcare services and the performance of health systems. External funding for NCDs, as a percentage of total external funding for health, was found to have a positive impact on the efficiency of NCDs spending while private domestic funding for NCDs was found to have a negative association with the efficiency of NCDs spending. The positive relationship between external funding for NCDs and efficiency of NCDs spending may be due to a number of reasons. Firstly, as noted by Atun et al. [75], external funding for NCDs can provide additional resources for NCDs prevention and treatment programs, which can enhance access to healthcare services and improve health outcomes. Secondly, external funds can also provide technical assistance and capacity building, and promote awareness about the importance of NCDs prevention and treatment, resulting in greater commitment and investment in NCDs programs from political leaders [76]. However, the negative relationship between private domestic funding for NCDs per capita and NCDs spending efficiency may convey mixed indications on the efficiency of NCDs spending. On one hand, domestic funding for NCDs improves access to NCDs healthcare services and enhances health outcomes, particularly in LLMCs where public health funding is limited [76]. On the other hand, it can indicate the financial burden households face when accessing NCDs healthcare services [77] if the private domestic funding is dominated by out-of-pocket payments, which could lead to inefficiencies in the healthcare system.

The analysis of the Malmquist total productivity index (MPI) which is a product of efficiency change and technical change indicate a decline which was largely driven by technical regress. There was an improvement in the efficiency change, which implies that health systems have moved closer to the production frontier. However, the decline in technical change eclipsed the growth in the efficiency change leading to a decline in the MPI. Over the five-year period that this study covers, all the 34 sampled SSA countries, with the exception of two (Central African Republic and DR Congo), registered a growth in efficiency change. This means that the use of NCDs resources in SSA region have improved between 2015 and 2019, it also indicates the need for further improvement. However, the technical change of all the countries was less than one, meaning technology (production) frontier has shifted downwards. Some previous studies on productivity in healthcare system obtained similar results [22, 78]. The decline in technology could be attributed to low adoption of new technologies in the treatment and prevention of NCDs in the region. Due to poor incentive arrangements within the healthcare systems in most SSA countries, healthcare providers prioritize the use of drugs and medical tests in managing NCDs over a more cost-effective treatment and preventive interventions [79]. Limited attention is paid to preventive interventions such as creating awareness for the need to reduce the intake of salt, alcohol, tobacco and increase physical activity. Again, the decline in technological change implies the need to increase investment in NCDs research and development in the region.

This study has strengths and limitations that are worth mentioning. For the strength of the paper, even though efficiency of health systems is widely studied, this is the first to provide evidence on the efficiency of NCDs spending in SSA. In terms of limitations, we acknowledge that despite the fact that all the data used in the study were obtained from credible and trustworthy sources, some of the data points were estimates which can affect the consistency of the

data leading to errors in the analysis and interpretation of the findings. Additionally, some countries were excluded from the study due to a significant amount of missing data, and including them in a future study could alter the estimates. Again, it is important to note that there might be a time lag between any investment in the input variables and its impact on the output variables. However, investigating such a time lag was beyond the scope of this study and is, therefore, acknowledged as a limitation. Nonetheless, the consistency of the estimates across the various models and robustness checks conducted in the study suggests that these limitations did not bias the findings.

## Conclusion

This study highlights the urgent necessity to enhance the efficiency and productivity of non-communicable disease (NCD) spending in Sub-Saharan Africa (SSA) to achieve Sustainable Development Goal 3.4 by the year 2030. Policymakers must prioritize the following areas to effectively address the challenge: (1) Increasing investment in NCD research and development to foster innovation and advancements in prevention, treatment, and management. (2) Implementing robust tobacco control policies, including raising tobacco taxes, enforcing smoke-free laws in public places, and providing incentives for smokers to quit, to curb the prevalence of tobacco-related NCDs. (3) Launching targeted education and awareness programs to promote healthier lifestyles and encourage households to adopt preventive measures against NCDs. (4) Investing in improving governance quality and strengthening health systems to ensure efficient allocation of resources and effective implementation of NCD interventions. (5) Expanding universal health coverage (UHC) initiatives to reduce out-of-pocket payments for NCD healthcare services, making them more accessible and affordable for all segments of society. By prioritizing these key areas, policymakers can make significant strides towards achieving SDG 3.4 and effectively tackle the growing burden of NCDs in SSA.

## Supporting information

**S1 Appendix. Descriptive statistics on all the variables.**
(DOCX)

**S2 Appendix. Bias-corrected technical efficiency scores of SSA countries (2015–2019).**
(DOCX)

**S3 Appendix. Variance inflation factor test for multicollinearity.**
(DOCX)

**S4 Appendix. Pair-wise Pearson's correlation coefficients.**
(DOCX)

**S5 Appendix. Tobit regression results.**
(DOCX)

**S6 Appendix. Descriptive statistics based on income groups (2015–2019 averages).**
(DOCX)

## Author Contributions

**Conceptualization:** Kwadwo Arhin, Disraeli Asante-Darko.

**Formal analysis:** Kwadwo Arhin, Disraeli Asante-Darko.

**Investigation:** Kwadwo Arhin.

**Methodology:** Kwadwo Arhin.

**Software:** Kwadwo Arhin, Disraeli Asante-Darko.

**Supervision:** Kwadwo Arhin.

**Visualization:** Kwadwo Arhin.

**Writing – original draft:** Kwadwo Arhin.

**Writing – review & editing:** Kwadwo Arhin, Disraeli Asante-Darko.

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
