## [Decision Letter · Decision Letter 0]

21 Jun 2023

PONE-D-23-12526Performance Evaluation of National Healthcare Systems in the Prevention and Treatment of Non-Communicable Diseases: Data Envelopment AnalysisPLOS ONE

Dear Dr. Arhin,

Thank you for submitting your manuscript to PLOS ONE. After careful consideration, we feel that it has merit but does not fully meet PLOS ONE’s publication criteria as it currently stands. Therefore, we invite you to submit a revised version of the manuscript that addresses the points raised during the review process.

A comprehensive and useful analysis by the authors are commendable with the use of multiple datasets covering a topic of high importance today. I hope the authors will be able to address to the comments by the reviewers to make this manuscript even stronger.

We look forward to receiving your revised manuscript.

Kind regards,

Krishna Kumar Aryal

Academic Editor

PLOS ONE

Journal Requirements:

3. Please remove your figures from within your manuscript file, leaving only the individual TIFF/EPS image files, uploaded separately. These will be automatically included in the reviewers’ PDF.

4. We note that Figure 1 in your submission contain map images which may be copyrighted. All PLOS content is published under the Creative Commons Attribution License (CC BY 4.0), which means that the manuscript, images, and Supporting Information files will be freely available online, and any third party is permitted to access, download, copy, distribute, and use these materials in any way, even commercially, with proper attribution. For these reasons, we cannot publish previously copyrighted maps or satellite images created using proprietary data, such as Google software (Google Maps, Street View, and Earth). For more information, see our copyright guidelines: http://journals.plos.org/plosone/s/licenses-and-copyright.

(1) You may seek permission from the original copyright holder of Figure 1 to publish the content specifically under the CC BY 4.0 license.  

**Additional Editor Comments:**

In addition to the comments from reviewers, please address the following.

Title: kindly include/mention that this analysis is for SSA in the title.

Please shift the strengths and limitations from the conclusion to the discussion. It is a struggle to find the key message from your analysis in the conclusion. Please try to have the most important message from the results of your analysis with possible recommendations.

Reviewers' comments:

Reviewer's Responses to Questions

**Comments to the Author**

1. Is the manuscript technically sound, and do the data support the conclusions?

Reviewer #1: Yes

Reviewer #2: Partly

Reviewer #3: Partly

2. Has the statistical analysis been performed appropriately and rigorously? 

Reviewer #1: I Don't Know

Reviewer #2: Yes

Reviewer #3: No

3. Have the authors made all data underlying the findings in their manuscript fully available?

Reviewer #1: No

Reviewer #2: Yes

Reviewer #3: No

4. Is the manuscript presented in an intelligible fashion and written in standard English?

Reviewer #1: Yes

Reviewer #2: Yes

Reviewer #3: Yes

5. Review Comments to the Author

Reviewer #1: • The last part of introduction section that details the contents in different sections can be removed.

• Sections are not numbered as per PLOS format and thus can be removed.

• Currently, methodology section spreads over 9.5 pages which could be shortened for the convenience of reader.

• Conclusion section of the study does not need to focus on does not need to summarize the methodology part. First paragraph of the conclusion is not needed.

• Together with description, provide the exact link from which data were extracted.

Reviewer #2: This is an interesting study measuring system performance across the relatively large number of countries using global database.

However, there are some limitations that I would suggest addressing:

1. Since two of the output variables selected in this study are high level impact indicators (mortality and DALY). Considering this, there will be reasonable time lag between the inputs and output variables. That means impact of increased budget and human resources will be reflected on mortality and DALY after certain time period and could sustain for multiple years. Therefore, due attention should be given on this, otherwise need to be mentioned as a major limitation.

2. Moreover, input variables selected in the study mainly cover recurrent costs (per capita expenditure and HR) and do not take into account the existing infrastructure which will have direct implication on efficiency. This is partially taken care by the variable returns to scale but inclusion of infrastructure such as number of hospital beds and/no number of primary health care centers could be considered. For enhancing the efficiency, right mix of the fixed (capital) and variable (recurrent) becomes an important aspect. Therefore, please consider dealing with capital aspects as well otherwise clarify on this.

3. In the discussion section, there is "financial burden households face when accessing NCDs healthcare services [67], which could lead to inefficiencies in the healthcare system." This does not seem to be a plausible statement or at least causal path is not straight forward.

Domestic private financing can be either in the form of prepaid fund e.g., in the form of health insurance premium or out of pocket payments. If private financing is mainly in the form of OOP, it can pose barrier on health service utilization and hence adversely affects the outputs variables such as service coverage and mortality. Otherwise, system efficiency may not be impacted just because who is paying for it.

Reviewer #3: The manuscript addresses an interesting issue. The analysed data collect a wide range of information and several features deserve to be properly modelled. The use of frontier methods is generally sound, though rather basic. Results are promising, but some checks are still required. Some comments follow.

1. The data are not fully available. This does not allow netiher for the reproduciblity of the results. Please, provide the data and the code used to obtain the results as a supplementary material. More details about the used software are required.

2. The statistical methods employed are rather basic. The same characteristics that make DEA a powerful tool can also create problems. The authors should keep these limitations in mind when choosing whether or not to use DEA. Just to mention a few: since DEA is an extreme point technique, noise (even symmetrical noise with zero mean) such as measurement error can cause significant problems; DEA is good at estimating "relative" efficiency of a DMU but it converges very slowly to "absolute" efficiency, it can tell you how well you are doing compared to your peers but not compared to a "theoretical maximum."; since DEA is a nonparametric technique, statistical hypothesis tests are difficult and are the focus of ongoing research; since a standard formulation of DEA creates a separate linear program for each DMU, large problems can be computationally intensive. Overall, the approache should be better motivated, as stochastic frontiers are often preferred to DEA.

3. I am not fully sure that such a basic approach is able to adequately capture all the data features. Please, refer to Panwar, A., Olfati, M., Pant, M. et al. A Review on the 40 Years of Existence of Data Envelopment Analysis Models: Historic Development and Current Trends. Arch Computat Methods Eng 29, 5397–5426 (2022). https://doi.org/10.1007/s11831-022-09770-3 and Omrani, H., Valipour, M., & Emrouznejad, A. (2021). A novel best worst method robust data envelopment analysis: incorporating decision makers’ preferences in an uncertain environment. Operations Research Perspectives, 8, 100184 for general overviews of extended approaches. I feel that it is rather important to investigate and properly account for atypical observations, noise, outliers, etc

4. I do not really get how the longitudinal and spatial structure of the data is modelled. Observations are not independent over time and space. These features are completely overlooked and may strongly affect the inferential results.

5. I am wondering why the analysis is commented assuming an a-priori endogenous clustering. My feeiling is that a clustering structure is present in the data, but it has to be estimated along with other model's parameters. The assumption that countries are homogeneous given the income is rather questionable.

6. Regression results should be further investigated. The assumptions underlying the regression modelling must be checked. Moreover, the p-value should be treaten with caution (please, describe how it is obtained from the bootstrapped estimates).

6. PLOS authors have the option to publish the peer review history of their article (what does this mean?). If published, this will include your full peer review and any attached files.

Reviewer #1: No

Reviewer #2: **Yes: **Ghanshyam Gautam

Reviewer #3: No

---

## [Author Response · Author response to Decision Letter 0]

18 Jul 2023

Editor’s Comments and Our Responses:

Comment 1. Please ensure that your manuscript meets PLOS ONE's style requirements, including those for file naming.

Response: We have made the necessary changes to the manuscript to ensure that our paper conforms to PLOS ONE’s style and policy. All the files have also been named accordingly.

Comment 2: In your Data Availability statement, you have not specified where the minimal data set underlying the results described in your manuscript can be found. PLOS defines a study's minimal data set as the underlying data used to reach the conclusions drawn in the manuscript and any additional data required to replicate the reported study findings in their entirety. All PLOS journals require that the minimal data set be made fully available.

Response: We have deposited the minimal data set underlying the results described in our manuscript as well as all the STATA 17 and DEAP 2.1 do files at Zenodo repository with URLs of https://zenodo.org/record/8151509 and DOI of https://doi.org/10.5281/zenodo.8151509 which makes it publicly available. The data set and the do files would make it easy for anyone to replicate the results of the study. Besides, we have added an Excel file of the data set and the STATA do files as Supporting Information files.

Comment 3: Please remove your figures from within your manuscript file, leaving only the individual TIFF/EPS image files, uploaded separately. These will be automatically included in the reviewers’ PDF.

Response: The figures have been saved and uploaded separately in TIFF image files.

Comment 4: Figure 1 in your submission contains map images that may be copyrighted. All PLOS content is published under the Creative Commons Attribution License (CC BY 4.0), which means that the manuscript, images, and Supporting Information files will be freely available online, and any third party is permitted to access, download, copy, distribute, and use these materials in any way, even commercially, with proper attribution. We cannot publish previously copyrighted maps or satellite images created using proprietary data, such as Google software (Google Maps, Street View, and Earth).

Response: To avoid infractions with copyright laws, we removed Figure 1 from the manuscript.

Comment 5: Please include captions for your Supporting Information files at the end of your manuscript, and update any in-text citations to match accordingly.

Response: Captions for all the Supporting Information files have been included at the end of the manuscript.

Additional Editor Comments: In addition to the comments from reviewers, please address the following.

Title: kindly include/mention that this analysis is for SSA in the title. Please shift the strengths and limitations from the conclusion to the discussion. It is a struggle to find the key message from your analysis in the conclusion. Please try to have the most important message from the results of your analysis with possible recommendations.

Response: The title has been modified to include sub-Saharan Africa (SSA). The strengths and limitations of the paper which were initially included in the conclusion section have been moved to the discussion section of the paper. The conclusion section has been rewritten focusing on the most important takeaways of the study and some recommendations have been provided. 

Reviewer #1 Comments and Our Responses:

Comment 1: The last part of the introduction section that details the contents in different sections can be removed.

Response: It has been removed.

Comment 2: Sections are not numbered as per PLOS format and thus can be removed.

Response: We have made the necessary changes to the manuscript to ensure that our paper conforms to PLOS ONE’s style and policy. The numbering of the sections has been removed.

Comment 3: Currently, the methodology section spreads over 9.5 pages which could be shortened for the convenience of the reader.

Response: After a thorough review of the methodology section, we have made some revisions to streamline the content and eliminate any unnecessary sub-sections and expressions. However, despite these efforts, we were unable to significantly reduce the number of pages as initially anticipated. It is crucial to ensure that the methodology section remains comprehensive and coherent to facilitate readers' understanding. Consequently, removing essential parts of the methodology may hinder readers' comprehension and their ability to grasp the study's approach.

Comment 4: The conclusion section of the study does not need to focus on does not need to summarize the methodology part. The first paragraph of the conclusion is not needed.

Response: The summary of the methodology in the conclusion section has been removed. The conclusion section has been rewritten focusing on the most important takeaways of the study. 

Comment 5: Together with the description, provide the exact link from which data were extracted.

Response: We made an effort to include the exact links from which the data were extracted. However, we acknowledged that doing so resulted in a longer methodology section, which contradicted the rationale of your comment 3. As a compromise, we cited each database from which the data were extracted within the manuscript. Additionally, we provided the exact links from which the data were extracted in the reference section to ensure transparency and proper referencing.

Reviewer #2 Comments and Our Responses:

Comment 1: Since two of the output variables selected in this study are high-level impact indicators (mortality and DALY). Considering this, there will be a reasonable time lag between the inputs and output variables. That means the impact of increased budget and human resources will be reflected in mortality and DALY after a certain time period and could sustain for multiple years. Therefore, due attention should be given to this, otherwise need to be mentioned as a major limitation.

Response: This has now been duly acknowledged as a limitation of the study in the manuscript.

Comment 2: Moreover, input variables selected in the study mainly cover recurrent costs (per capita expenditure and HR) and do not take into account the existing infrastructure which will have direct implications on efficiency. This is partially taken care of by the variable returns to scale but the inclusion of infrastructure such as a number of hospital beds and/or the number of primary health care centers could be considered. For enhancing efficiency, the right mix of fixed (capital) and variable (recurrent) becomes an important aspect. Therefore, please consider dealing with capital aspects as well otherwise clarify this.

Response: We appreciate the important role capital or healthcare infrastructure plays in the provision of healthcare services and including it as one of the inputs is the ideal. However, the unavailability of data to measure capital stock in the healthcare sector makes it difficult to include it in empirical studies. The number of hospital beds per 1000 people (i.e. hospital bed density) is usually used as a proxy for capital stock in healthcare production studies. However, this data is simply not reported for many years in most sub-Saharan African (SSA) countries. As a result, most DEA studies, particularly those that utilize panel data that span over a number of years as is the case for this study, use healthcare expenditure per capita (HEPC) as the main input variable. 

Comment 3: In the discussion section, there is a "financial burden that households face when accessing NCDs healthcare services [67], which could lead to inefficiencies in the healthcare system." This does not seem to be a plausible statement or at least the causal path is not straightforward. Domestic private financing can be either in the form of prepaid funds e.g., in the form of health insurance premiums or out-of-pocket payments. If private financing is mainly in the form of OOP, it can pose a barrier to health service utilization and hence adversely affect the outputs variables such as service coverage and mortality. Otherwise, system efficiency may not be impacted just because of who is paying for it.

Response: That statement has been revised in the manuscript to make it clearer. We thank Reviewer #2 for these comments.

Reviewer #3 Comments and Our Responses:

Comment 1: The data are not fully available. This allows neither the reproducibility of the results. Please, provide the data and the code used to obtain the results as supplementary material. More details about the used software are required.

Response: We have deposited the minimal data set underlying the results described in our manuscript as well as all the STATA 15 and DEAP 2.1 do files at Zenodo repository with URLs of https://zenodo.org/record/8151509 and DOI of https://doi.org/10.5281/zenodo.8151509 which makes it publicly available. The data set and the do files would make it easy for anyone to replicate the results of the study. Besides, we have added an Excel file of the data set and the STATA do files as part of the supplementary materials. Details of the Stata and DEAP 2.1 software used in the estimations have been provided in the manuscript.

Comment 2: The statistical methods employed are rather basic. The same characteristics that make DEA a powerful tool can also create problems. The authors should keep these limitations in mind when choosing whether or not to use DEA. Just to mention a few: since DEA is an extreme point technique, noise (even symmetrical noise with zero mean) such as measurement error can cause significant problems; DEA is good at estimating the "relative" efficiency of a DMU but it converges very slowly to "absolute" efficiency, it can tell you how well you are doing compared to your peers but not compared to a "theoretical maximum."; since DEA is a nonparametric technique, statistical hypothesis tests are difficult and are the focus of ongoing research; since a standard formulation of DEA creates a separate linear program for each DMU, large problems can be computationally intensive. Overall, the approach should be better motivated, as stochastic frontiers are often preferred to DEA.

Response: We agree with Reviewer #3 about some limitations of the DEA approach in measuring efficiency. These limitations have been duly acknowledged in the discussion section of the paper. However, DEA is used in this study because it has some advantages over the stochastic frontier analysis (SFA). These include its ability to handle multiple input and output variables and require no a priori specification of the production function. We have explained in the manuscript that these advantages motivated the use of the DEA approach.

Comment 3: I am not fully sure that such a basic approach is able to adequately capture all the data features. Please, refer to Panwar, A., Olfati, M., Pant, M. et al. A Review on the 40 Years of Existence of Data Envelopment Analysis Models: Historic Development and Current Trends. Arch Computat Methods Eng 29, 5397–5426 (2022). https://doi.org/10.1007/ s11831-022-09770-3 and Omrani, H., Valipour, M., & Emrouznejad, A. (2021). A novel best worst method robust data envelopment analysis: incorporating decision makers’ preferences in an uncertain environment. Operations Research Perspectives, 8, 100184 for general overviews of extended approaches. I feel that it is rather important to investigate and properly account for atypical observations, noise, outliers, etc.

Response: We thank Reviewer #3 for the references provided. They have really improved our knowledge of DEA. To ensure the robustness of the results and investigate the potential impact of atypical observations and outliers, we conducted robustness checks using different combinations of input and output variables and by excluding countries considered to be potential outliers to test the base model. The closeness of the mean efficiency scores from all the different scenarios indicates the stability of the base model used in this study. Again, we used an alternative model, Tobit regression, to test the robustness of the bootstrap regression employed to examine the association between the estimated efficiency scores and the environmental variables. The high degree of similarity of the results compared with that of the bootstrap regression in terms of coefficient values, signs, and statistical significance is an indication of the robustness of the empirical evidence given in this study. 

Comment 4: I do not really get how the longitudinal and spatial structure of the data is modeled. Observations are not independent over time and space. These features are completely overlooked and may affect the inferential results.

Response: We appreciate the concern of Reviewer #3 on this issue. However, in data envelopment analysis (DEA), the approach differs from stochastic frontier analysis (SFA) in terms of assumptions and underlying concepts. In SFA, it is common to assume that the observed data are affected by random noise and error terms, and the goal is to estimate a production frontier that separates efficient and inefficient units. On the other hand, DEA is a non-parametric method that does not assume specific functional forms or stochastic noise. It aims to identify the most efficient units by comparing their input-output performance relative to other units in the dataset. Besides, the bootstrap regression used in the second stage analysis provides robust estimates by repeatedly resampling the data and calculating efficiency scores. This resampling helps account for variations in the data and provides a more reliable estimation of efficiency, especially when the dataset contains outliers or influential observations.

Comment 5: I am wondering why the analysis is commented on assuming an a priori endogenous clustering. My feeling is that a clustering structure is present in the data, but it has to be estimated along with other models’ parameters. The assumption that countries are homogeneous given the income is rather questionable. 

Response: We investigated the impact of the potential presence of these issues conducting robustness checks using different combinations of input and output variables and by excluding countries considered to be potential outliers given their levels to test the base model. The closeness of the mean efficiency scores from all the different scenarios indicates the stability of the base model used in this study and that the issues raise do not significantly affect the results of the study.

Comment 6: Regression results should be further investigated. The assumptions underlying the regression modeling must be checked. Moreover, the p-value should be treated with caution (please, describe how it is obtained from the bootstrapped estimates).

Response: We investigated the estimates from the bootstrap regression by using an alternative model, Tobit regression, to test their robustness. The high degree of similarity of the estimates compared with that of the bootstrap regression in terms of coefficient values, signs, and statistical significance is an indication of the robustness of the empirical results from the bootstrap regression.

---

## [Decision Letter · Decision Letter 1]

7 Aug 2023

PONE-D-23-12526R1Performance evaluation of national healthcare systems in the prevention and treatment of non-communicable diseases in sub-Saharan Africa: Data envelopment analysisPLOS ONE

Dear Dr. Arhin,

Thank you for submitting your manuscript to PLOS ONE. After careful consideration, we feel that it has merit but does not fully meet PLOS ONE’s publication criteria as it currently stands. Therefore, we invite you to submit a revised version of the manuscript that addresses the points raised during the review process.

Kindly find some remaining comments from Reviewer 3

We look forward to receiving your revised manuscript.

Kind regards,

Krishna Kumar Aryal

Academic Editor

PLOS ONE

Additional Editor Comments:

Dear Authors, majority of the responses have been found satisfactory by the reviewers. However, you are kindly requested to address the remaining comments from Reviewer 3.

Reviewers' comments:

Reviewer's Responses to Questions

**Comments to the Author**

1. If the authors have adequately addressed your comments raised in a previous round of review and you feel that this manuscript is now acceptable for publication, you may indicate that here to bypass the “Comments to the Author” section, enter your conflict of interest statement in the “Confidential to Editor” section, and submit your "Accept" recommendation.

Reviewer #2: All comments have been addressed

Reviewer #3: (No Response)

2. Is the manuscript technically sound, and do the data support the conclusions?

Reviewer #2: Yes

Reviewer #3: Partly

3. Has the statistical analysis been performed appropriately and rigorously? 

Reviewer #2: Yes

Reviewer #3: No

4. Have the authors made all data underlying the findings in their manuscript fully available?

Reviewer #2: Yes

Reviewer #3: Yes

5. Is the manuscript presented in an intelligible fashion and written in standard English?

Reviewer #2: Yes

Reviewer #3: Yes

6. Review Comments to the Author

Reviewer #2: (No Response)

Reviewer #3: Thank you very much for all your replies to my former comments.

Nevertheless, there is still a general lack of statistical knoweldge.

1. Although pros and cons of the proposed approach are discussed properly, no solutions to the discussed drawbacks are proposed, applied, investigated. Mainly, statistical inference is a crucial aspect of any empirical analyses. A comparison with other approaches, SFA for example, is required.

2. Longitudinal and spatial dimensions do not refer to noise. Both must be included, as a main assumption of the modelling (i.e. independence across observations) is not tenable. You can easily include both even in the DEA modelling, there are several papers discussiong both in the literature.

3. Please, provide evidence that results are robust against endogenous clustering and/or other a-priori exogenous clustering.

7. PLOS authors have the option to publish the peer review history of their article (what does this mean?). If published, this will include your full peer review and any attached files.

Reviewer #2: No

Reviewer #3: No

---

## [Author Response · Author response to Decision Letter 1]

23 Oct 2023

Reviewer #3 Comments and Our Responses:

Comment 1: Thank you very much for all your replies to my former comments. Nevertheless, there is still a general lack of statistical knowledge. Although pros and cons of the proposed approach are discussed properly, no solutions to the discussed drawbacks are proposed, applied, investigated. Mainly, statistical inference is a crucial aspect of any empirical analyses. A comparison with other approaches, SFA for example, is required.

Response: We have conducted a stochastic frontier analysis (SFA) using the same dataset and compared the results with the data envelopment analysis (DEA) approach that was initially used (see pages 7, 8, 19, 21, and 22). 

Comment 2: Longitudinal and spatial dimensions do not refer to noise. Both must be included, as a main assumption of the modelling (i.e., independence across observations) is not tenable. You can easily include both even in the DEA modelling, there are several papers discussing both in the literature.

Response: We have conducted a rigorous spatial autocorrelation analysis examine any spatial agglomeration of NCDs spending efficiency across the sub-Saharan Africa. Longitudinal analyses were also carried in the revised manuscript (see pages 10 – 12, 24 – 26, Tables 8 & 10, and Figure 4).

Comment 3: Please, provide evidence that results are robust against endogenous clustering and/or other a-priori exogenous clustering. 

Response: We applied the cluster-robust standard errors, clustering at the country level, in the estimation of the SFA model to correct for any potential heteroscedasticity problem in the data (see pages 20 and Table 6). We want to thank Reviewer #3 for his/her insightful comments. It took us more than two months to learn many new things to be able to satisfy all the comments, but it is worth the effort. We, therefore, acknowledge his/her contribution to this work.

Thank you.

---

## [Decision Letter · Decision Letter 2]

7 Nov 2023

Performance evaluation of national healthcare systems in the prevention and treatment of non-communicable diseases in sub-Saharan Africa

PONE-D-23-12526R2

Dear Dr. Arhin,

We’re pleased to inform you that your manuscript has been judged scientifically suitable for publication and will be formally accepted for publication once it meets all outstanding technical requirements.

Kind regards,

Krishna Kumar Aryal, MPH, PhD

Academic Editor

PLOS ONE

Additional Editor Comments (optional):

Thank you for addressing all the comments raised by the reviewers and the editor.

Reviewers' comments:

Reviewer's Responses to Questions

**Comments to the Author**

1. If the authors have adequately addressed your comments raised in a previous round of review and you feel that this manuscript is now acceptable for publication, you may indicate that here to bypass the “Comments to the Author” section, enter your conflict of interest statement in the “Confidential to Editor” section, and submit your "Accept" recommendation.

Reviewer #3: All comments have been addressed

2. Is the manuscript technically sound, and do the data support the conclusions?

Reviewer #3: (No Response)

3. Has the statistical analysis been performed appropriately and rigorously? 

Reviewer #3: (No Response)

4. Have the authors made all data underlying the findings in their manuscript fully available?

Reviewer #3: (No Response)

5. Is the manuscript presented in an intelligible fashion and written in standard English?

Reviewer #3: (No Response)

6. Review Comments to the Author

Reviewer #3: (No Response)

7. PLOS authors have the option to publish the peer review history of their article (what does this mean?). If published, this will include your full peer review and any attached files.

Reviewer #3: No

---

## [Editor Report · Acceptance letter]

8 Nov 2023

PONE-D-23-12526R2 

Performance evaluation of national healthcare systems in the prevention and treatment of non-communicable diseases in sub-Saharan Africa 

Dear Dr. Arhin:

I'm pleased to inform you that your manuscript has been deemed suitable for publication in PLOS ONE. Congratulations! Your manuscript is now with our production department. 

Kind regards, 

on behalf of

Dr Krishna Kumar Aryal 

Academic Editor

PLOS ONE